Speeding up iterative applications of the BUILD supertree algorithm

Redelings Benjamin D. 1 2 3
Holder Mark T. mtholder@ku.edu 3 4
1 Biology Department, Duke University , Durham , NC , United States of America
2 Ronin Institute , Durham , NC , United States of America
3 Ecology and Evolutionary Biology, University of Kansas , Lawrence , KS , United States of America
4 Biodiversity Institute, University of Kansas , Lawrence , KS , United States of America
Gillespie Joseph
Electronic publication date: 2024 Jan 2
Publication date: 2024
Volume: 12
Electronic Location ID: e16624
Received 2023 Mar 3; Accepted 2023 Nov 16
Copyright: 2024 Redelings and Holder
Copyright year: 2024
Copyright holder: Redelings and Holder
License: This is an open access article distributed under the terms of the Creative Commons Attribution License, which permits unrestricted use, distribution, reproduction and adaptation in any medium and for any purpose provided that it is properly attributed. For attribution, the original author(s), title, publication source (PeerJ) and either DOI or URL of the article must be cited.
License URL: https://creativecommons.org/licenses/by/4.0/

Keywords: Supertree, Build algorithm, Optimization, Phylogenetics

Funding: The National Science Foundation of the United States of America (Division of Biological Infrastructure ) 1759838 This work was supported by the National Science Foundation of the United States of America (Division of Biological Infrastructure Award No. 1759838). The funders had no role in study design, data collection and analysis, decision to publish, or preparation of the manuscript.

==============================
The Open Tree of Life (OToL) project produces a supertree that summarizes phylogenetic knowledge from tree estimates published in the primary literature. The supertree construction algorithm iteratively calls Aho’s Build algorithm thousands of times in order to assess the compatability of different phylogenetic groupings. We describe an incrementalized version of the Build algorithm that is able to share work between successive calls to Build. We provide details that allow a programmer to implement the incremental algorithm BuildInc, including pseudo-code and a description of data structures. We assess the effect of BuildInc on our supertree algorithm by analyzing simulated data and by analyzing a supertree problem taken from the OpenTree 13.4 synthesis tree. We find that BuildInc provides up to 550-fold speedup for our supertree algorithm.

Introduction

The Open Tree of Life (OToL) project summarizes phylogenetic knowledge from tree estimates published in the primary literature. Curators for the project import published trees, associate the tip labels of the trees to standardized taxonomic labels, and correct errors in trees that occurred during the deposition of the trees into repositories. OToL also produces a synthesis tree (Hinchliff et al., 2015) that combines hundreds of input phylogenies with a comprehensive taxonomic tree from the Open Tree Taxonomy (Rees & Cranston, 2017, OTT hereafter). This “synthetic tree” is a supertree—a tree that is produced by combining multiple input trees, and having a leaf label-set that is the union of the leaf label-sets of the input trees (Gordon, 1986).

Our supertree method is intended to summarize and transparently represent the published input trees, not to produce a phylogeny estimate that is more accurate than the inputs (Redelings & Holder, 2017). Each edge of the supertree corresponds to a supporting branch in one of the input trees. The synthetic tree can be used as a comprehensive phylogeny of living and extinct taxa. It can also be used as a means of navigating the OToL curated collection of published input trees, and of exploring conflict between them. The OToL portal at https://tree.opentreeoflife.org allows browsing or downloading the latest release of the synthetic tree. It also allows for uploading and curating input phylogenies.

The OToL synthetic tree is created by an algorithm that will add a grouping from an input tree to the full tree if that grouping is compatible with the previously added groups. More specifically, the supertree algorithm iterates over branches of input phylogenies to determine if the groups subtending each internal branch can be added to the synthesis tree (Redelings & Holder, 2017). The order in which input phylogenies are considered is an input to the algorithm, and this order is provided by data curators for the OToL project. The taxonomy tree is always the last phylogeny to be considered.

The current implementation of the supertree algorithm is fast enough to allow the full supertree to be updated periodically. A faster implementation would allow users to explore the effects of differing inputs, such as a different set of phylogenies or a different ranking of trees. This would enable constructing alternative synthetic trees on demand via the web-interface for a variety of users.

The core algorithm for determining compatibility of each potential grouping with previously added groups is the classic Build (Aho et al., 1981) algorithm. Build is invoked thousands of times during the construction of the supertree using OToL’s pipeline (Redelings & Holder, 2017). Each invocation applies Build to the set of all previously added groups (which are already known to be consistent) plus one new group. Here we describe improvements to naively calling Build iteratively. Implementing the incrementalized BuildInc algorithm has resulted in dramatic reductions in running times for the key steps in the supertree pipeline. This new algorithm will allow the OToL project to offer more frequent updates to the synthetic tree and explore features such as on-demand supertree construction under the direct control of users of the project’s web-services.

While we focus on the use of Build within the OToL project, Build is an ingredient that is used in a wide range of algorithms, such as computing a consensus of equally likely trees (Sanderson, McMahon & Steel, 2011), inferring species trees from gene trees (Roch & Warnow, 2015), determining orthology and paralogy relations between genes in a gene family (Lafond & El-Mabrouk, 2014), and hierarchical clustering (Chatziafratis, Niazadeh & Charikar, 2018). In order to determine what opportunities are opened up by an incrementalized version of Build, researchers must examine each algorithm that uses Build. However, we highlight two possiblities. First, an incrementalized Build might be used to construct an online version of an existing algorithm—that is, an algorithm in which input is added in batches and that produces a complete result after each batch. A batch could represent a new data point, or it could represent the next gene in a genome-wide scan. Second, an incrementalized Build might aid in discovering a compatible subset of trees or bi-partitions instead of merely declaring failure when a given set is incompatible. Thus, an incrementalized version of Build has implications beyond the OToL project.

Context for Build in the current OToL pipeline

The current pipeline divides the full supertree problem into subproblems as described in Redelings & Holder (2017). These subproblems cover regions of the taxonomy tree in such a way that adjacent sub-problems contain common nodes (higher taxa) but do not contain common edges. Each subproblem also contains the relevant region of any input tree that coincides with the region of the taxonomy tree. Repeated calls to Build produce a supertree solution to each subproblem. Solving the subproblems is a time-consuming step for the entire pipeline. Increasing the speed of the subproblem-solver would speed up the pipeline, but would also allow it to handle larger subproblems. Larger subproblems occur when OToL curators add more input phylogenies to the pipeline and this large set of inputs conflict with more groupings found in the taxonomy. Larger subproblems also occur when the final location of an incertae sedis taxon is far from its initial location in the taxonomy (see Redelings & Holder, 2019 for a discussion of the handling of incertae sedis taxa). The increased speed might eliminate the need for decomposing the supertree problem into subproblems.

Methods

We will begin the methods description with a review of the Build algorithm and the description of the new optimizations to the algorithm that are the focus of this paper. After the algorithmic discussion, the simulation study to evaluate performance will be explained.

Overview of the Build algorithm

The Build algorithm is a recursive algorithm that determines if a set of rooted triplets of the form x, y|•z are jointly compatible (Aho et al., 1981). Here, we use the symbol • to stand for the root of the tree to emphasize the rooting of the triple. If they are compatible it constructs a tree that displays all the triplets.

The algorithm works by creating a graph for each recursive level. If the graph forms a single connected component, then the Build algorithm has detected an incompatibility between the set of triplets. The graph contains a node corresponding to each leaf that is relevant to the current recursion level. At the root-most level of the recursion every leaf is relevant. A rooted triple is relevant to a recursion level only if each of the three leaves is relevant at that level. For each relevant triple x, y|•z an edge is added between x and y. Since the leaves in the more closely related pair {x, y} are sometimes called the “cluster” of the triple, the resulting graph is called the “cluster graph” for the given set of triplets.

If no triplets are incompatible on a level, then the procedure applies Build to each connected component of the cluster graph. This is the recursive aspect of the algorithm. The nodes that form a connected component at one level will constitute the relevant nodes for the next recursive call. Thus the relevant leaf set is partitioned at each level. However, the edges between nodes are not passed to the next recursive level. Instead they are constructed from the set of relevant triplets at each level. A rooted triple that is relevant at one level may not be relevant at a subsequent recursion level. Thus, not all rooted triplets have a corresponding edge at the next level. A set of triplets is jointly compatible only if the Build algorithm finds no incompatibility at any recursive level.

Rooted splits

The Build algorithm was originally designed to work on sets of rooted triplets. However, when working with evolutionary trees, it is more convenient to work with sets of rooted splits because each branch corresponds to a rooted split. A rooted split σ on taxon set T is written σ = σ1|•σ2, where σ1 and σ2 are non-overlapping subsets of T. We refer to σ1 as the include group or “cluster”, and σ2 as the exclude group. The root of the tree is on the side of the exclude group. Note that σ1∪σ2 may be smaller than T.

It is straightforward to extend Build to operate on a set of rooted splits by treating each rooted split as shorthand for all the rooted triplets that it implies. So Build is commonly modified to take a set of rooted splits Σ instead of a set of rooted triplets.

The relationship of the current optimizations to previous work

Previous approaches to speed up Build have tried to improve the order of computation for the analysis at each recursive level. Deng & Fernández-Baca (2018) decreased the order to OMlog2M for a set of phylogenies with M = (number of edges) + (number of leaves). This is accomplished by decreasing the time for the analysis on each recursive level from OM2 to Olog2M. It also involves sharing work between different levels of the recursive algorithm by retaining a graph between successive levels instead of creating a new graph from scratch on each level.

Our approach differs from previous work because we attempt to share work between successive calls to Build. When calling Build with one set of splits followed by calling Build again with one additional split, we seek to reuse work from the first call while performing the second. This requires saving work from the first call in an object that represents the solution to Build, and passing that solution object as input to the second call. We do not attempt to decrease the order of the computation as a function of the number of leaves or edges. Instead we base our incremental approach on a naive algorithm that has total cost OM3. It may be possible to create an algorithm that is both incremental and has a more favorable order of computation for large M, but we do not attempt that here.

Data structures used to explain implementations of Build

In order to explain our incremental Build algorithm in an understandable manner, we seek to introduce the full complexity of the algorithm in stages.1 We thus begin by describing a version of the traditional Build algorithm that saves its work in a solution data structure. This will allow us to focus on changes to the algorithm instead of changes to the data structures when we introduce the first incremental version of the algorithm at a later stage. Our version of the Build algorithm constructs the connected components of the cluster graph without explicitly constructing the cluster graph. That is, our algorithm does not directly represent the edges of the cluster graph in memory.

We begin by introducing the Solution and Component record types that we will use to store temporary work during the algorithm. Using the terminology common to object-oriented programming, we will refer to an instance of these record types residing in a computer’s memory as an “object.” The following sections will describe the distinct steps in Build using some common programming notational conventions: (1) parentheses after the name of an algorithm denote the arguments supplied to that algorithm and (2) the period (or “dot”) notation after the name of a data object is used to refer to a field within that record.

An initial Solution object can be created before applying the Build algorithm. Upon termination, the result of the algorithm will be emitted by storing it in that Solution object. As mentioned above in the overview of the algorithm, each level of Build’s recursion starts with a set of taxa and a set of relevant splits. Connected components of taxa are created and merged as each input split is considered. In our object-oriented description of the algorithm, this corresponds to creation of a Solution object for the current level of recursion. That Solution object will hold a set of Component objects –each of which will store information about the connected components created during the algorithm. Each Component represents a sub-problem that needs to be solved. If no incompatibility is detected at the current level, then the algorithm recursively calls Build to create a Solution object for each Component.

Thus the Solution object forms a tree by indirect recursion to mirror the recursive structure of Build: each Solution object can contain some Component objects, and each Component object contains a Solution object (see Section B.2 of the Supplementary Information File). We refer to the tree associated with a Solution object as a “solution data structure”.

The solution and component objects for a level of Build

The definition of the Solution and Component record types is given in Fig. 1. Creating a new Solution object is done with a procedure referred to as CreateBlankSolution(T), where T is a collection of taxon identifiers. A new solution object, S, has only its S.T field initialized; That field holds a copy of the taxon identifiers; thus initialization has computational complexity O|T|.

Figure 1 Definitions for the Solution and Component record types.

The field C.O is not used by non-incremental Build.

Defining the Build algorithm

To start the algorithm, an initial Solution object can be created and initialized by filling its S.T field with the complete taxon set, T. All other fields of the solution object are initially empty. We can think of the entire Build algorithm as taking a taxon set T and set of rooted splits, Σ. The full Build algorithm consists of the following steps:

1. initialize a Solution object, S to contain the taxon set T. We will refer to this set of operation as a function: CreateBlankSolution(T);

2. call a helper algorithm BuildA (S, Σ); and

3. return the result stored in object S.

The BuildA algorithm describes the set of operations to be performed at a single level of recursion as well as how to call the next levels of recursion.

Dissecting the operations required at each level of BuildA

The operations performed in BuildA (S, Σ) can be separated in several steps:

1. RemoveIrrelevantSplits(S, Σ) to remove splits from Σ that are not relevant at the current level. Splits that are removed here as irrelevant are also implied by the split S.T| •(T-S.I) and are recorded on the solution in field S.I.

2. MergeComponents(S, Σ) to find the connected components of the cluster graph by merging any two components that overlap the include group (“cluster”) of a split in Σ.

3. Fail(S). Return Failure if there is only one connected component.

4. AssignSplitsToComponents(S, Σ) to the single component on the next recursive level where they may be relevant.

5. Iterate. For each non-trivial component C ∈S.C:

(a) use CreateBlankSolution(C.T) to create a new solution object for each Component object and store this in C.S.

(b) Recurse. Call BuildA (C.S, C.ΔΣ) on the sub-problem for each non-trivial component C, Return Failure if that call fails.

The RemoveIrrelevantSplits, MergeComponents, and AssignSplitsToComponents steps all have the form of a for-loop that iterates over splits in Σ and modifies S. This fact will be used later, when we seek to incrementalize these steps. It suggests a naive incrementalization strategy of simply iterating over any newly added splits ΔΣ to add their effects to S.

The RemoveIrrelevantSplits(S, Σ) step

We assume that each split in Σ has an include group fully contained in S.T. This means that a split is relevant at this recursive level if and only if its exclude set intersects the relevant taxon set (S.T).

This procedure examines each split σ in Σ. If σ is not relevant, then σ is added to the collection S.I of implied splits and removed from Σ. Splits removed in this step were relevant at previous recursion levels, but are irrelevant for the solution at or below the recursive level S because they do not separate any taxa in S.T from other taxa in S.T. Determining which splits to remove has computational complexity OV×E where V = |T| and E = |Σ|.

Although this is not central to the operation of the algorithm, we note the splits in S.I are implied by the branch of the solution tree leading to S. Thus the solution tree implies all the splits in Σ, and split σ is recorded on the branch that implies it.

It is not necessary to perform this step at the root level of the recursion, as all tips are inside of T for that solution object.

The MergeComponents(S, Σ) step

The MergeComponents step partitions taxa according to which connected component of the cluster graph they are in. This step can also be thought of as partitioning taxa in T according to an equivalence relation, where taxa are equivalent if they are in the same connected component of the cluster graph. Note in this version of Build we construct the connected components of the cluster graph without constructing the graph.

To perform the partitioning, we begin by assigning each taxon t ∈ T to its own connected component {t}. This corresponds to a graph with no edges. We then consider each split σ ∈ Σ and merge any components that overlap σ1 into a single connected component. This is equivalent to adding edges to the cluster graph connecting all taxa in σ1.

Our implementation represents the components as two related maps between components and elements: (1) the linked list of taxon identifiers included in each component (the C.T field of the component C), and (2) the array that maps each taxon to its component (stored in S.M).

Detecting mergers requires considering every taxon in every include group, and has order O|Σ|×|T|. There can be at most |T| mergers. The cost of merging linked lists is just O|T| since merging linked lists is an O1 operation. Unfortunately, when we merge two components C1 and C2 where C2 is smaller, we must also rewrite the array entry for taxa in C2. The cost per taxon is the number of times that taxon is part of a merger where it is in the smaller component. Since this can happen up to log2|T| times per taxon in the worst case,2 the cost is |T|log|T|. The total cost is thus O|Σ|×|T|+|T|log|T|. If we define M = |Σ| + |T| following Deng & Fernández-Baca (2018), then this is OM2+MlogM=OM2.

The Fail(S) step

Fail can be done in O1 by having an array S.C of active non-trivial components, and checking the size of that array. If the size is 1, and the taxon set for the single component has the same size as S.T, then the operation fails.

The AssignSplitsToComponents(S, Σ) step

For each split σ ∈ Σ we have a guarantee that all the taxa in σ1 are in the same component by definition of the cluster graph. We refer to the component that contains the include group of σ as its “corresponding component”. We can determine the corresponding component of any split σ by simply looking up the component for the first element of σ1 in S.M. We implement the assignment of a split σ to a component C by placing a reference to σ into C.ΔΣ.

Simple optimization: trivial and non-trivial components

In a solution object S, we store an array S.M of pointers to components. We also implement S.T as an array, so that if t = T[i] then taxon t belongs to component S.M [i].

Trivial components are defined as components that contain only one taxon. An optimization to improve memory usage and speed is to simply use a Null reference in the S.M array to indicate that the corresponding taxon is in a trivial component, rather than creating a component object for every trivial component. This requires only minor tweaks to the MergeComponents step to create a new Component object on-the-fly whenever a set of trivial components are merged into each other with no non-trivial component involved in the merger.

Example #1: creating a solution

Consider calling Build T,Σ with taxa T=A1,A2,B and splits Σ = {A1A2|•B}. This creates a solution data structure shown in Fig. 2. Build begins by creating a Solution object S1 where S1.T =[A1, A2, B] and then calling BuildA (S1, Σ).

Figure 2 The Solution object S1 returned by calling Build([A1, A2, B], {A1A2|•B}).

Solution objects have rounded edges and a pink border. Component objects have rectangular edges and a blue border. The temporary values C1.ΔΣ and C1.O are shown with the values they contain before being cleared. Solid arrows indicate pointers. Dashed arrows show the correspondence between Solution objects and nodes on the solution tree.

The BuildA creates a single non-trivial component C1 containing {A1, A2} and a single trivial component containing B. BuildA does not fail, because there is more than one component. The single split A1A2|•B is assigned to C1. BuildA then creates a new solution C1.S where C1.S.T = [A1, A2] and then calls BuildA (C1.S, C1.ΔΣ).

The second-level BuildA removes the split A1A2|•B from C1.ΔΣ and adds it to C1.S.I. It then calls MergeComponents, Fail, and AssignSplitsToComponents with no effects. No non-trivial components are created, and the call to BuildA succeeds because it contains the trivial components {A1} and {A2}. Since there are no non-trivial components, another round of recursion is not performed.

Stepwise construction of BuildInc

The goal of the incrementalized algorithm is to reuse previous work from computing Build T,Σ when computing Build T,Σ+ΔΣ. The incrementalized algorithm has the signature BuildInc (S, ΔΣ). Although the taxon set T and the previously added splits Σ are not explicitly present in the signature of BuildInc, they are contained within S. S may also contain intermediate results from a previous BuildInc call, so that BuildInc can reuse previous work. If BuildInc (S, ΔΣ) succeeds, then S is modified in-place to contain the new splits ΔΣ in addition to Σ. However, if BuildInc fails, then S is unmodified.

In order to simplify the description of the incrementalized algorithm BuildInc, we first introduce two partially-incrementalized algorithms BuildInc′ and BuildInc″. This allows us to simplify the explanation of BuildInc by introducing concepts in a step-wise fashion. The BuildInc′ and BuildInc″ algorithms both mutate the solution object even when the algorithm returns False. The fully incrementalized algorithm BuildInc adds the ability to track changes that are made to the solution object, and then roll them back if the algorithm ultimately returns False. We now describe key difference in the steps of the first incrementalized algorithm BuildInc′. We will use the ′and ″symbols to decorate the names of algorithms based on whether they are used in BuildInc′ or BuildInc″.

The incrementalized algorithm allows adding input splits either one-at-a-time, or in larger batches. We assume that the input splits have been partitioned into a series of non-overlapping subsets ΔΣi according to some strategy that is chosen in advance. Each subset constitutes a batch. The simplest strategy is to set ΔΣi = {σi} for some split σi, so that splits are added one-at-a-time. However, it is possible to incorporate input splits into the solution object in larger batches, and we explore this option in the Results section below.

Given a partition of the input splits, the general structure for all of the incrementalized algorithms is:

1. initialize a Solution object, S, to contain the taxon set T. We will refer to this set of operation as a function: CreateBlankSolution(T);

2. BuildIncA (S, 0̸)

3. for each subset ΔΣi

(a) call BuildIncA (S, ΔΣi)

4. return the result stored in object S.

In the descriptions that follow, Σ will be used to denote the set of splits that have been added in previous increments of BuildInc′ without failing. Thus, Σ + ΔΣ would be the set of splits included if the current round of BuildInc′ succeeds, and ΔΣ is the set of splits that are new to the current increment of BuildInc′.

BuildIncA′ (S, ΔΣ): RemoveIrrelevantSplits ′(S, ΔΣ) step

This is identical to RemoveIrrelevantSplits except that (for reasons described below) S.I may already contain some splits before this step begins.

BuildIncA′: MergeComponents′ (S, ΔΣ) step

We seek to construct the connected components of the cluster graph for the splits Σ(S) + ΔΣ from the connected components for Σ(S). We will refer to connected components under the cluster graph for Σ(S) as “original” components. The addition of the splits ΔΣ may add edges to the cluster graph, but it cannot remove any. Therefore, the addition of ΔΣ may merge original components, but it cannot split them. In order to compute the new connected components, we simply need to iterate over splits in ΔΣ and continue merging components.

Characterizing how the components for Σ + ΔΣ are related to any original components is central to our approach. The non-trivial components for Σ + ΔΣ can be characterized as new, modified, or unmodified.

• new (a component composed entirely of previously-trivial components)

• unmodified (a non-trivial component present in the original S.C)

• modified (a non-trivial component that contains at least one original non-trivial component)

If S is initially empty, then all non-trivial components will be new. If S is not initially empty, then all three types of non-trivial components can occur. If an original component C′ is a subset of a component C, then we say that C′ is subsumed by C. We now describe how these three classes of non-trivial components retain original solutions.

A new (non-trivial) component C will have an empty solution C.S. There is no original solution to retain, since all its subsumed components are trivial.

For unmodified (non-trivial) components, we retain the single original Solution object in the field C.S. We will modify the retained solution if any splits from ΔΣ are assigned to C.

Modified (non-trivial) components have a taxonomic set C.T that was not a connected component in the previous iteration of BuildIncA′. However, the Component object does not need to be created de novo; instead, we repurpose the Component object from one of the subsumed non-trivial components to represent the new, larger equivalence class. The previous solution in C.S is not the solution for the enlarged taxon set. However, we retain all the solutions for all subsumed non-trivial components in an additional field for in the component data structure, C.O.

BuildIncA′: AssignSplitsToComponents ′(S, ΔΣ) step

Splits in ΔΣ must be assigned to their corresponding components by placing them in C.ΔΣ, just as in BuildIncA. However, splits in Σ are treated differently depending on which original component they were previously assigned to.

Splits from Σ(S) that were assigned to an unmodified component C are still associated with that component. This is because they were contained in C.S, and it has been retained.

Splits from Σ(S) that were assigned to a modified component C are no longer associated with C because C.S has been discarded. We therefore iterate over original components C′ that were subsumed by C and append their splits to C.ΔΣ (see Section B.2 of the Supplementary Information File).

New components cannot contain any splits from Σ(S), so we do not need to do anything extra for them.

BuildIncA′: recursive call to BuildIncA′ step

We seek to construct a solution for each of the components C ∈ S.C using BuildIncA′. In all cases we do this by calling BuildIncA′ (C.S, C.ΔΣ) to add splits in C.ΔΣ to the solution C.S. For unmodified components, C.S is the original solution for C, so previous work is re-used. Any corresponding splits from ΔΣ are added to this original solution.

For modified components, C.S is a Null reference indicating that no solution has been calculated. We construct an empty Solution with taxa C.T. The set C.ΔΣ contains splits corresponding to C from both Σ(S) and ΔΣ. Therefore, despite the existence of previous work in C.O, we do not manage to re-use any of it.

For new components, C.S is also Null. We also construct an empty Solution with taxa C.T. The set C.ΔΣ contains only splits from ΔΣ. There is no previous work that could be re-used here.

Example #2: adding a split to an unmodified component

When incrementally adding a split that resolves a node below the top level, BuildIncA′ passes the split down the tree until it reaches the level of the resolved node. At levels higher than the resolved node, the split is assigned to one of the components, but does not modify it. At the resolved node, the split either modifies components or causes the creation of a new non-trivial component.

To see this, let us consider incrementally adding the split σ2 = a1a2|•a3 to a Solution object that contains σ1 = a1a2a3|•b (Fig. 3). Here the split σ2 resolves a node in the original solution data structure.

Figure 3 Adding a split to an unmodified component.

Top: the result of adding the split A1A2A3|•B. Bottom: the result of incrementally adding the split A1A2|•A3, starting with the top Solution.

Running BuildIncA′ (S1, {σ2} leaves the only non-trivial Component C1 at the top level unchanged. σ2 is Assign ed to C1 and ends up in C1.ΔΣ.

This leads to a call to BuildIncA′ (S2, {σ2}). σ2 is not removed, so S2.I is unchanged. However, σ2 leads to the creation of a new non-trivial component C2 in Merge. σ2 is Assign ed to C2 and ends up in C2.Σ.

Finally, we get a call to BuildIncA′ (S3, {σ3}). Here σ2 is finally removed, and placed into S3.I. No non-trivial components are created, so we are done.

Example #3: merging two components

When incrementally adding a split that groups two children of a node in the solution tree, the components that correspond to the grouped children tend to re-emerge at a lower level. However, BuildIncA′ is not able to recognize this, and so must solve each of these re-emergent components from scratch.

To see this, let us consider incrementally adding the split a1b1|•c to a Solution object that contains a1a2|•b1 and b1b2|•c (Fig. 4). The new split merges the original components C1 (containing and {A1, A2}) and C2 (containing {B1, B2}). The new component {A1, A2, B1, B2} is then stored in the modified Component object for C1.

Figure 4 Merging two components.

(A) The initial solution contains the splits A1A2|•B1 and B1B2|•C. In both (B) and (C), when A1B1|•C is incrementally added, the original components C1 and C2 are merged and stored in the modified Component object for C1.The original component {A1, A2} re-emerges at a lower level as C3. (B) In BuildIncA′, C1.ΔΣ contains all three splits, and not just the incrementally added split. The re-emergent component {A1, A2} is solved from scratch. (C) In BuildIncA″, C1.ΔΣ contains only the new split A1B2|•C. The original solution S3 is punctured, and its split B1B2|•C is added to S4.I. However, the original solution S2 for {A1, A2} is passed down to C3 and can be re-used, saving work.

The BuildIncA handles merged components by extracting the splits from subsumed solutions. Therefore, C1.ΔΣ contains all three splits, and not just the incrementally added split. We can see that the original component {A1, A2} re-emerges at a lower level as C3. However, BuildIncA′ solves it from scratch. The component {B1, B2} does not re-emerge at a lower level. This is because the split b1b2|•c can be satisfied on the same level, and is not passed down.

The partially incrementalized algorithm BuildInc″

The BuildInc′ algorithm can only reuse work for unmodified components; modified components must be recomputed from scratch. Figuring out how to re-use previous work for modified components was one of the most difficult steps in designing BuildInc. The key insight is that a solution data structure can be regarded as a collection of splits (see section ??). Thus, instead of extracting the splits from original solutions that were subsumed by a modified component, we may simply pass down the original solutions themselves.

Performing operations such as MergeComponents, AssignSplitsToComponents, and RemoveIrrelevantSplits turns out to be more efficient when the input splits are packaged in original solutions. These operations frequently do not break apart the collections of splits.

Most importantly, passing down original solutions can sometimes allow us to reuse solutions instead of recomputing them from scratch. When an original component C′ is subsumed by merging, it may sometimes re-emerge deeper in the recursion stack as a descendant of the merged component. If the original solution S manages to percolate down the tree to the new location of C′ then we can re-use S. However, if any of the splits in Σ(S) is satisfied at an earlier recursion level, then S must be broken up and cannot be re-used. This will be described in further detail below and illustrated in Example #4.

We therefore modify the signature of BuildIncA′ (S, ΔΣ) to BuildIncA″ (S, ΔΣ, O), where O is a set of original solutions. The call BuildIncA′ (S, ΔΣ, O) indicates an attempt to add both splits in ΔΣ and splits in O to the splits in S. We will write Σ(O) to indicate the splits contained in O, where ΣO= ⋃S′∈OΣS′

These original solutions in O satisfy three properties that we will prove in Section A of the Supplementary Information File:

1. If O is non-empty, then S contains no splits.

2. Each original solution in O has a taxon set that is a subset of S.T.

3. All the original solutions in O have non-overlapping taxon sets.

BuildIncA″: MergeComponents ″(S, ΔΣ, O) step

We seek to construct the connected components of the cluster graph for the splits Σ(S)∪ΔΣ∪Σ(O) from the connected components for Σ(S). In addition to iterating over ΔΣ and merging components, BuildIncA″ must additionally handle splits in Σ(O). One possible approach would be to iterate over splits in Σ(O) and merge components that overlap each split. However, it turns out that there is a more efficient way.

We note that each original solution S′ ∈ O is a connected component of the original cluster graph for some higher recursive level. Additionally, the taxa in S′.T are connected solely by edges corresponding to splits in Σ(S′) (see Section ??). This is because any split whose include group overlaps the component is assigned to (and only relevant to) that component. Therefore, the effect of the splits Σ(S′) is only to connect the taxa in S′.T. We may therefore iterate over original solutions S′ ∈ O and merge any two components that both overlap S′.T. This makes it unnecessary to extract the splits σ ∈ S′.

BuildIncA″: AssignSplitsToComponents ″(S, ΔΣ, O) step

The BuildIncA″ needs to assign splits in Σ(S), ΔΣ, and Σ(O) to their corresponding components. Each split in ΔΣ is assigned to a component C and stored in C.ΔΣ, just as as in BuildA and BuildIncA′. However, there are two major differences from BuildIncA′.

First, BuildIncA″ does not need to do anything for splits in Σ(S), because they are already assigned to their correct component. BuildIncA′ needed to extract splits from C.O and add them to ΔΣ if C was a merged component. However, BuildIncA′ passes C.O down to the next recursive level directly, so this is unnecessary. Note that since BuildIncA″ no longer assigns splits from Σ(S) to C.ΔΣ, C.ΔΣ will consist only of splits from ΔΣ.

Second, BuildIncA′ must assign splits in Σ(O) that were recieved from the previous recursive level to their corresponding component. However, all splits for a subsolution S′ ∈O have to end up in the same component. This is because S′.T must be entirely contained in one component. The splits of S′ must then be in the same component. Therefore, we may simply assign original solutions S′ to components in their entirety. We can determine which component a solution S′ goes to by looking up the component for any element in S′.T.

BuildIncA″: recursive call to BuildIncA″

As before, BuildIncA″ (S, ΔΣ, O) must iterate over the non-trivial components C ∈S.C and construct a solution C.S for each of them. In all cases this is done by calling BuildIncA″ (C.S, C.ΔΣ, C.O). Passing original solutions from modified components down the tree in C.O allows us to reuse previous work, as described in the section on RemoveIrrelevantSplits ′ below. The only difference from BuildIncA′ is the third argument, the set of original solutions, O.

BuildIncA″: RemoveIrrelevantSplits ″(S, ΔΣ, O) step

There are two differences between BuildIncA′ (S, ΔΣ) and BuildIncA″ (S, ΔΣ, O). First, O might contain a solution S′ to the problem we are trying to solve. In that case, we would like to reuse that solution. Second, when checking for implied splits, we need to check inside each original solution S′ ∈ O as well as inside ΔΣ. We now discuss these differences in greater detail.

Reusing previous work in BuildIncA″ (S, ΔΣ, O)

Suppose that O contains a solution S′ with the same taxon set as S. In that case, we would like to re-use the previous work in S′ instead of solving the problem from scratch.

We first note that Σ(S) must be empty, since O was non-empty, so it is safe to discard S and use S′ instead. Second, O is empty after removing S′. This is because S′ contains all the taxa in S.T and there are no remaining taxa in S.T for any other solutions to contain.

We therefore replace the call to BuildIncA″ (S, ΔΣ, O) with BuildIncA″ (S′, ΔΣ, 0̸). This preserves the invariant that S and O cannot both be non-empty.

Removing implied splits from original solutions

Recall that a split σ should be placed into S.I if and only if σ2 does not intersect S.T. For an original solution S′ ∈ O, we claim that only splits in S′.I can satisfy this condition. If a split σ from S′ is not in S′.I, then σ2 must intersect S′.T. But S′.T is a subset of S.T, so σ2 intersects S.T as well and cannot go into S.I. Therefore, for each original solution S′, we only need to check the splits in S′.I.

If any split in S′.I meets the criterion, then we remove it from S′.I (conceptually) and move it to S.I. However, if one of the splits is removed from a solution, then the solution no longer has the property that its splits are all in the same connected component. It is no longer a solution. In such a case, we say that the solution is “punctured”.

In order to retain our invariants, we remove punctured solutions from O. However, we must retain the splits that they contain. For each punctured original solution S′, we copy the splits in S′.I that were not moved to S.I into the set ΔΣ. We move the child solutions S′. Ci.S into O. In this way, all the splits of S′ are retained—some in S.I, some in ΔΣ, and some in O.

Original solutions in O that are not punctured may be retained unmodified.

Note that replacing an original solution S′ ∈O with its child solutions retains the invariants that solutions in O (i) have non-overlapping taxon sets and (ii) are contained within S.T. The taxon sets of child solutions to a solution S′ are contained within S′.T and are non-overlapping. Since S′.T is contained within S.T and does not overlap with any other solutions in O, its child solutions must also be contained within S.T and not overlap any other solutions in O.

Example #4: merging two components and re-using original solutions

When incrementally adding a split that groups two children of a node, the components that correspond to the grouped children tend to re-emerge at a lower level. While BuildIncA′ solves the solutions from scratch, BuildIncA″ is able to re-use the original solutions to these components.

To see this, we show how BuildIncA″ solves the same problem that BuildIncA′ solved in Example #3 (Fig. 4). Unlike BuildIncA′, BuildIncA″ leads to C1.ΔΣ containing only the single incrementally-added split A1B1|•C. Instead of extracting the splits A1A2|•B1 and B1B2|•C from the original solutions S2 and S3, BuildIncA′ passes down the original solutions themselves.

The original component {A1, A2} re-emerges at a lowever level, and is solved by re-using the original solution S2. The other original solution S 3 is punctured, and its split B1B2|•A is added to S4.I.

The fully incrementalized algorithm BuildInc

The BuildInc′ and BuildInc′ algorithms modify the original solution data structure as they execute. When these algorithms discover that the additional splits are incompatible with the previous solution, we cannot simply revert to the previous solution, because it has been modified. We must therefore recreate the previous solution data structure from scratch, discarding all of the saved work.

We address this problem by extending BuildInc″ to record any change that it makes to the original solution. We can then reverse these changes in the case of failure. We call the extended algorithm BuildInc. Much of the description of BuildInc thus boils down to (i) specifying just what information must be recorded to reverse changes made by BuildIncA″ and (ii) some optimizations that avoid spending time recording information that will never be used.

An overview of the rollback approach

All modifications to each Solution object S are complete before any modifications are made to its children. We can therefore represent modifications to the solution data structure as a sequence R of modifications to individual Solution objects. If BuildIncA returns failure at the top level, we can then walk this sequence in reverse order, and roll back the changes that it describes. We create the record type RollbackInfo to record modifications made to an individual Solution object (Fig. 5).

Figure 5 Definitions for the RollbackInfo and MergeRollbackInfo.

Component mergers are the most interesting type of modifications that BuildIncA′ makes to Solution objects. We can represent component mergers for each Solution object as a sequence of individual mergers of two components. For each component merger, we record enough information about the two original components to reverse the merger. It is then possible to reverse the mergers by walking the sequence in reverse order, and undoing each merger in turn. We create the record type MergeRollbackInfo to record modifications made to an individual Solution object (Fig. 5).

Mergers are of two types: mergers of two non-trivial components, or mergers of a non-trivial component with a trivial component. We handle mergers of two trivial components, by creating an empty “non-trivial component” data structure, and merging it with each trivial component in turn.

Details of rollback info

The RollbackInfo record type

Changes that occur to S during BuildIncA (S, ΔΣ) include (i) appending additional implied splits to S.I, (ii) modifying the component list S.C, and (iii) merging components. We can therefore record the changes that occur to S in terms of (i) the original set of implied splits S.I, (ii) the original list of components S.C, and (iii) a sequence of records that describe individual merges of two components. We note that implied splits are only ever appended to the end of S.I. Therefore, as an additional optimization we can simply record the original length of S.I, and revert to the original version by truncating the array to that length.

The MergeRollbackInfo record type

When merging two non-trivial components C1 and C2, let us assume that C2 is the smaller component. The modifications that occur to C1, C2, and S are the following:

• S.M [t] is set to C1 for each taxon t ∈C2.T.

• the elements of C2.T are removed and appended to C1.T.

• C1.S is set to Null.

We can restore C1.S from the saved reference. The pointer to the first element of C2.T allows us to split the linked list C1.T in two at the proper place, and return the latter half to C2.T. We can then walk the restored elements of C2.T and set S.M [t]=C2 for each t ∈C2.T.

Sometimes we merge a non-trivial component C with a trivial component containing the taxon t. In such cases, there is no non-trivial component C2. We indicate such cases in the RollbackInfo object by setting component2 = Null. Such mergers can be rolled back by removing the last element of C1.T, setting S.M[t] = Null and setting C1.S = orig_solution.

Optimizations

Recording and replaying rollback info allows us to avoid discarding saved work. However, both recording and replaying rollback info also have a cost. In order to achieve the optimum speedup from rollback info, we must avoid paying this cost when we do not need to.

Optimization #1

We only need to undo changes to a Solution object if it was part of the previous solution data structure. In order to determine if a Solution object S is part of the previous solution data structure, we initialize a counter S.visits to 0 when creating a new Solution object. We then increment the counter each time the Solution object is visited by BuildIncA. We then avoid appending the rollback info for S to the sequence of changes R if S.visits =0.

Optimization #2

Sometimes an original Solution object contains only trivial components. In such a case, we do not need to walk the list of merge records in reverse, undoing each component merger. We can simply clear the component list, and set all the entries of S.M to Null.

Optimization #3

If the number of original components is 0, then we will either not record the rollback info r at all (optimization #1), or we will record r but not look at the merge records (optimization #2). In that case, creating the sequence of merge records is a waste of time. We therefore pass a flag to MergeComponentWithTrivial and MergeComponents indicating whether or not to record merge records.

This optimization is essential because it avoids creating merge records for cases where they will not be used. One of those cases is when implementing Build (Σ) by calling BuildIncA (S, Σ) for a blank Solution S. In order for BuildInc not to be slower than Build, we must avoid creating merge records in this case.

Modifications to BuildIncA

In order to record rollback info, we must make a few modifications to BuildIncA. In RemoveIrrelevantSplits, we record the original number of implied splits. In MergeComponents we record (i) the original number of components, (ii) a merge-record for each component merger, and (iii) a copy of C after new components are added, but before empty components are removed.

Rollback (S,r)

After running BuildIncA, BuildInc must run RollbackAll (S, R) in case of failure. This consists of running RollbackOne (S, r) on individual RollbackInfo objects r.

1. Truncate S.I to its previous length r.n_orig_implied_splits.

2. If r.n_orig_components is equal to 0, then clear S. C and set S.M [t] = Null for each taxon t.

3. If r.n_orig_components is more than 0, walk the list of merge records in reverse order, and undo each one.

4. If we recorded the original components vector S0.C then

(a) swap(S.C, S0.C).

(b) Truncate S.C to r.n_orig_components.

This process seems simple enough, but one aspect of it that is tricky. Some components are created during merging that (i) are not original components, and also (ii) end up being empty. So they are not final components. We need these components to survive (i.e not be deallocated) so that we can temporarily add elements to them during rollback. We will then move these elements out of them into original components.

Batching + Oracle

Batching

Recall that our supertree algorithm works by considering an ordered list of trees. We seek to construct the set of splits from these trees that are jointly compatible.

The batching approach works by batch-adding the splits for the each tree in order. To batch-add a group of splits, we try and add the whole group of splits. If that succeeds, then we keep the whole group. If it fails, and the group has one split then we are done. If it fails, and the group has more than one split, then we batch-add the first half of the group, followed by the second half of the group. This ends up being both simpler and more efficient than trying fixed-size batches because we do not need to figure out the ideal batch size and it has exponential back-off.

Batching improves efficiency when Σ is large because the cost of determining the compatibility of Σ + ΔΣ does not increase much as the size of ΔΣ increases. If all the splits in ΔΣ will be accepted, it is thus substantially more efficient to add them in one batch.

Oracle

The oracle first runs conflict analysis on each input phylogeny T to identify branches of T that conflict with the tree of currently accepted splits. We then batch-add splits corresponding to the non-conflicting branches of T. This makes batching more efficient by making it more likely that large batches do not contain any conflicting splits.

Unfortunately, the oracle cannot filter splits for the taxonomy tree if there are any incertae sedis taxa. This is because taxonomy branches may correspond to partial splits in the presence of incertae sedis taxa. Our current conflict analysis does not handle partial splits.

Simulation experiments

The simulations script (gen_subproblem.py) can be found in the otcetera repository on GitHub (https://github.com/OpenTreeOfLife/otcetera and via DOI: https://zenodo.org/doi/10.5281/zenodo.10041275). The user of the script specifies: (a) a number of leaves in the full tree, (b) a number of phylogenetic input trees to simulate, (c) a tip inclusion probability for each phylogenetic input, (d) a number of edge-contract-refine (ECR) moves to conduct on each input tree, and (e) an edge contraction probability for the taxonomy. For each replicate, the script uses DendroPy (Sukumaran & Holder, 2010) to generate a pure birth (Yule) tree with the specified number of leaves as the true(model) full tree for that replicate. The specified number of phylogenetic inputs are created by sub-sampling the full tree (using the tip-inclusion probability to assess whether a tip remains in the sampled input); if all tips are deleted a new phylogenetic input is drawn, rather than emitting an empty tree. Then the specified number of ECR moves are applied to the tree to mimic phylogenetic estimation noise. The last tree emitted for each replicate is designed to mimic the taxonomic input in the problems used by the Open Tree of Life project. The taxonomic input is complete (lacks any sub-sampling based on taxon inclusion probabilities). In addition to having errors introduced by ECR moves, the taxonomic input undergoes branch collapsing (using the user-supplied edge-contraction probability on each internal edge independently) to mimic the unresolved character of most taxonomies.

We simulated a collection of supertree problems containing 50-1000 taxa, all with 20 phylo-inputs.

Each OTU was included in the phylogeny inputs with probability 0.5. Each edge in the taxonomy was collapsed with probability 0.75. We introduced two ECR errors per tree. For each simulation condition, we determined the run time by averaging across 15 simulated data sets.

Results

We examined the effect of different optimizations by looking at their run time on simulated data sets and one real data set. Run times are for an Intel i7-5820K CPU with 32 Gb RAM running Linux.

Simulated data

We first examined the effect of the batching and oracle optimizations on simulated data sets as the number of taxa increased (Fig. 6). Batching yields a speedup that increases from 1.6-fold at 50 taxa to 17-fold at 1,000 taxa. Using the oracle to eliminate inconsistent splits shows no speedup when not paired with batching.

Figure 6 Effect of oracle and batch optimizations on run time.

The left panel is log-scaled, whereas the right panel is not. Run time versus number of OTUs where each sample data set has 20 phylogenetic trees as inputs.

However, when combined with batching, the oracle yields an additional speedup that increases from 1.3-fold at 50 taxa to 2.4-fold at 1,000 taxa. This indicates that the oracle allows larger batch sizes to succeed.

Given that our simulation protocol performs two ECR edits to each phylogeny, we expect about four splits to be inconsistent with the underlying tree per phylo-input. Therefore larger trees have a smaller fraction of inconsistent splits. That may explain why larger trees recieve a bigger speedup from batching.

Incrementalizing Build yields a larger speedup than the oracle+batch optimization (Fig. 7). The speedup increases from 20-fold at 50 taxa to 398-fold at 1000 taxa. This represents an additional 9.9-fold speedup over batching + oracle at 1000 taxa. Much of this speedup relies on the abilty to save work via rollback: the incremental algorithm achieves only a 116-fold speedup at 1,000 taxa. Rollback ability thus provides a speedup of 3.4-fold for BuildInc over BuildInc″.

Figure 7 Comparison of incremental BUILD and batch+oracle optimizations.

The left panel is log-scaled, whereas the right panel is not. Run time versus number of OTUs where each sample data set has 20 phylogenetic trees as inputs.

It is possible to combine the batching and oracle optimizations with BuildInc. This is because BuildInc allows ΔΣ to include a batch of splits instead of just adding one split at a time. Adding batching to BuildInc yields a slight speedup of 27% over BuildInc alone at 1,000 taxa. However adding batching + oracle yields a 2.1-fold slowdown.

OpenTree data

We also examined a data set taken from the OToL synthesis release 13.4. As mentioned above, the OToL project normally divides the full supertree problem into subproblems after taxonomy-only taxa are removed (Redelings & Holder, 2017). Here we consider the effect of optimizations on running the full supertree problem without dividing it. This is a much larger scale than the simulated data sets, which are designed to be similar to a single subproblem.

The data set includes 1,223 non-empty trees in addition to the taxonomy. The taxonomy tree has 94,028 leaves. To give an idea of the size of the input trees, the three largest input trees contain 11,217, 8,369, and 7,160 leaves respectively. All but 80 trees contain fewer than 331 leaves.

The total run time without optimization is 3,323 min = 55 h 23 min (Table 1). Batching + oracle decreases the runtime to 345 min, which is a 9.6-fold decrease. BuildInc decreases the runtime further to 5 min 44s. This is a 60-fold speedup over batching + oracle, and a 579-fold speedup overall. Surprisingly, if we disable rollback, the runtime is 527 min, which is even slower than batching + oracle. This indicates that the OToL supertree problem contains more conflicting splits than the simulated data set above. Additionally, BuildInc + batching + oracle achieves an additional 1.95-fold speedup over BuildInc. This is different than in the simulations above, where BuildInc + batching + oracle was slower than BuildInc.

Table 1 Run times for handling the OToL 13.4 data set without subproblem decomposition.

Batch	Oracle	Incremental	Rollback	Time	
0	0	0	0	3,323 m	
1	1	0	0	345 m	
0	0	1	0	527 m	
1	1	1	0	271 m	
0	0	1	1	5 m 44 s	
1	1	1	1	2 m 56 s	

When using the naive Build algorithm, our supertree algorithm considers a total of 183,850 splits and calls Build 183,850 times. Of these splits, 160,682 are accepted. So, 87.4% of splits are accepted and 12.6% are rejected.

When the oracle is enabled, 162,517 splits are considered. This lowers the fraction of rejected splits to 1.1%. When using the oracle + batch optimizations, Build is called only 14,481 times.

Figure 8 shows that some phylo-inputs take a lot more time than others. Processing splits from the taxonomy tree takes a large fraction of the total time. However, when graphed against the number of accepted splits, the relationship is more linear, indicating that the effect of large trees is at least partly driven by the fact that large trees have a large number of splits.

Figure 8 Time taken versus number of input splits accepted (top) or input trees processed (bottom).

The left panel is log-scaled, whereas the right panel is not.

Discussion

We set out to improve the speed of our supertree algorithm that calls Build many times in a row. By using an incremental algorithm that re-uses work from previous calls to Build we were able to achieve a speedup of up to 579-fold in practice.

Incrementalizing faster version of Build

Our current approach uses a naive approach to Build that achieves OM3 time. Future research should consider whether it is possible to modify our incremental algorithm to incorporate recent improvements to Build that decrease the order, such as Deng & Fernández-Baca (2018). In fact, their algorithm relies on another incremental algorithm –incremental graph connectivity–which identifies new connected components that appear as edges are deleted or added (Holm, de Lichtenberg & Thorup, 2001). One approach might therefore be to replace our map of taxa to Component objects at each level with a graph that supports incremental connectivity queries. This would involve incrementally adding any required edges at each recursive level and identifying connected components that are merged.

One difficulty with this strategy is that we assume the data structures used to find connected components on each recursive level are separate. In constrast, Deng & Fernández-Baca (2018) construct a single graph at the beginning of the algorithm, and then remove additional edges from the single, shared graph at each recursive level. It might be possible to adapt this approach by saving a copy of the relevant portion of the graph at each recursive level to preserve a snapshot of previous work that is unmodified by work at deeper levels. However, even if this is feasible, it is not yet clear how much this would change the running time of the algorithm.

Supplemental Information

Supplemental Information 1 Runtimes for various settings on 20 phylogenetic trees

Comparison of incremental BUILDINC with different problem sizes and optimizations enabled on simulated inputs. Tab-separated values format

Click here for additional data file.

Supplemental Information 2 Running times on 1223 subproblems with no optimizations enabled

Running times on 1223 subproblems with no optimizations enabled. Tab-separated values.

Click here for additional data file.

Supplemental Information 3 Running times on 1223 problems with only incremental optimization enabled

Running times on 1223 problems with only incremental optimization enabled. Tab-separated values.

Click here for additional data file.

Supplemental Information 4 Running times on 1223 problems with incremental and rollback optimizations enabled

Running times on 1223 problems with incremental and rollback optimizations enabled. Tab-separated values.

Click here for additional data file.

Supplemental Information 5 Running times on 1,223 problems with batch and oracle optimizations enabled

Running times on 1,223 problems with batch and oracle optimizations enabled. Tab-separated values.

Click here for additional data file.

Supplemental Information 6 Running times on 1,223 problems with batch, oracle, incremental, and rollback optimizations enabled

Running times on 1,223 problems with batch, oracle, incremental, and rollback optimizations enabled. Tab-separated values.

Click here for additional data file.

Supplemental Information 7 Source code

A snapshot of otcetera code base with SHA 2287cefc4d12d328c4d9a4272b10aa25b38abbcd from https://github.com/OpenTreeOfLife/otcetera. The algorithms described in this article are implemented in the otc-solve-subproblem tool.

Click here for additional data file.

Supplemental Information 8 Appendix

Click here for additional data file.

Additional Information and Declarations

Competing Interests

Author Contributions

Data Availability

1 One might even say “incrementally”.

2 The worst case is when you have a perfectly balanced merger tree for the components.

The authors declare there are no competing interests.

Benjamin D. Redelings conceived and designed the experiments, performed the experiments, analyzed the data, prepared figures and/or tables, authored or reviewed drafts of the article, and approved the final draft.

Mark T. Holder conceived and designed the experiments, performed the experiments, authored or reviewed drafts of the article, and approved the final draft.

The following information was supplied regarding data availability:

The data for the timing results depicted in figures are available in the Supplemental Files.

The otcetera code is available in the Supplemental File and at Zenodo: Redelings, B., & Holder, M. (2023). otcetera commit 2287cefc4d12d328c4d9a4272b10aa25b38abbcd. Zenodo. https://doi.org/10.5281/zenodo.10041276.

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
