# Peer review of "Speeding up iterative applications of the BUILD supertree algorithm"

_PeerJ, doi:10.7717/peerj.16624_

## Round 0.1 · original submission · Major Revisions

Dear Dr. Redelings and Holder:

Thanks for submitting your manuscript to PeerJ. I have now received two independent reviews of your work, and as you will see, the reviewers raised some concerns about the research. Despite this, these reviewers are optimistic about your work and the potential impact it will have on research studying methods for phylogeny estimation. Thus, I encourage you to revise your manuscript, accordingly, taking into account all of the concerns raised by both reviewers.

While the concerns of the reviewers are relatively minor, this is a major revision to ensure that the original reviewers have a chance to evaluate your responses to their concerns. There are many suggestions, which I am sure will greatly improve your manuscript once addressed.

Please make sure your workflow is reproducible. Make all scripts public and ensure they work. All figures should be legible and drawn to scale.

Therefore, I am recommending that you revise your manuscript, accordingly, taking into account all of the issues raised by the reviewers.

Good luck with your revision,

-joe

Reviewer 1 ·

Basic reporting

The paper describes an optimisation to the BUILD algorithm within the scope of the Open Tree of Life (OToL) project.
Overall, I find the paper well-written, easy to follow and of interest to the journal's readership.
I have to say that I especially enjoyed the footnote in page 3/20.

I do, however, have a few comments regarding the Introduction and Discussion sections:

1) No link is given to the OToL.
While the vast majority of the potential readership will already be familiar with the OToL, for others it may be the first time they encounter the project.

2) Given that a definition of supertree was given in lines 28-30, I think that Gordon 1986. J.Classif or an equivalent publication should be cited.
While BUILD is the earliest (or at least one of) algorithm adopted/developed for supertree building, the first definition of a consensus supertree (now usually called just supertree) I am aware of is in Gordon's paper.
There may be earlier instances, but I have not come across them.

3) Following from 1), it may also be advisable to make it clear that the OToL supertree is a synthesis of available phylogenetic knowledge, it is not concerned with supertree inference.
I know this is discussed in Redelings and Holder 2017 PeerJ, but a quick mention/reminder would not go amiss.

4) While a potential impact/usage of the newly described algorithm is mentioned in lines 49-50, I would welcome a slightly more in depth discussion of how this might influence future evolutionary studies that are not focused on systematics.
For a bioinformatics/algorithm focused journal the existing discussion would be more than suficient, but (even with the selection of the 'Bioinformatics and Genomics' section) for a journal with a readership with broader interests I think that the potential impact of the optimised algorithm on evolutionary studies could be expanded on.

This is otherwise a very clear and convincing article, with an easy to follow introduction/description of the original and modified BUILD algorithms, with informative and illustrative figures, and clear description of the methods and results.

Experimental design

As mentioned in the 'Basic Reporting', the text is clear throughout, including the explanation of how the simulated trees were obtained, and the discussion of the results and how/why they differ between the simulated and 'empirical' datasets.

I did, however, have some difficulty in understanding the in-graph legends for figs.6-8, particularly due to 'none' appearing multiple times in the same set of in-graph legends.
Slightly more descriptive in-graph legends might be advisable, and would make it easier to interpret all three sets of graphs.

As for the raw data, the unedited tables of the runtimes results are all available, but the tree sets used as input are not.
For reproducibility purposes, I believe the tree sets should be added to the raw data already supplied (despite the generating script being available, but see point 4 below), along with the details of the system/machine used for the analyses.
The latter need not be added to the main text, mentioning them somewhere in the supplementary files would be sufficient.

The source code is very clearly annotated and I was able to follow it easily, despite not programming with C++.
However, the source code/installation instructions provided need a few small tweaks to make the installation process easier (the code was compiled in a virtual machine running Ubuntu 20.04.4):
1) The restbed installation writes the library to a folder called 'library', not 'lib' as specified in the otcetera build files

2) In otcetera meson searches for libclucene, which is not listed as a dependency.
Installation proceeds without issue without this library, but its absence is still flagged.
(Also, library was installed but it was not found - assuming ineptitude on my part here.)

3) ninja installation tests needed 'usr/bin/env/python' to pass all tests, but given lack of reference to Python in installation instructions it is unclear whether Python 2.7 was required or Python 3.X installed with python-is-python3.
All tests passed after installation of Python 2.7.

4) The gen_subproblem.py script needed to generate the simulated data is available from the otcetera repository in the OpenTreeOfLife GitHub account (and the source code provided in the supplementary materials), but not in the mtholder account that is listed in the installation instructions.

Validity of the findings

See 'Basic Reporting' and 'Experimental Design' for comments on reproducibility of the analyses and discussion of the impact of this work.

Additional comments

Despite clarity of the writing there are some minor details and/or typos that need the authors' attention:

Line 54: Maybe end 'These subproblems...' sentence with 'regions of interest' or 'regions in the synthetic tree'
The way the sentence is written can be confusing if the reader is not already familiar with BUILD.
Given that it seems this paragraph's aim is to give a crash course on BUILD in the context of the OToL, the ambiguity is counter productive, despite the extremely clear overview of BUILD in the section that follows.

Line 68: Double checking if authors mean to use triples or triplets, as both are accepted terms in this context.

Line 77: 'call' should be 'called'

Line 114: 'verson' should be 'version'

Line 132: 'of Component' remove 'of'

Line 213: There are two periods after 'graph'

Line 251: Do you mean 'the key difference' or 'key differences'?

Line 258: The 'until that collection is exhausted:' in point 3 of the pseudocode is confusing...
I interpret it as step 3.(a) is called iteratively until all 'DeltaSigmas' have been analysed, but am not confident of this interpretation.

Line 418: Is the lower case 'd' in 'REMOVEd' intentional?

Line 424: Believe the dash is supposed to be an m-dash, here looks like a hyphen (which can be confused with a minus sign)

Line 529: 'that are (i) are' remove one of the 'are'

Line 547: 'that of conflict' remove 'of'

Line 549: Is 'The' supposed to be 'This'?

Line 551: Comma after 'Unfortunately,

Line 555: Add the URL to one of the otcetera repositories?

Line 557: '(c) an tip' should be '(c) a tip'

Line 572: '2 ECR errors' unsure whether this should be 'two ECR errors', if ECR considered a unit author guidelines are technically being followed.
Otherwise, numbers 1-9 should be spelled out.
See also '4 splits' in line 585 for the same issue, other instances may be present.

Line 572: I would put a comma after 'condition'.

Line 598: Abbreviation of Open Tree of Life not used consistently.
In line 23 set as OToL, here written as OpenTree.

Line 606: Might be a good idea to show the conversion of 3323 minutes to hours, in order to really drive home the optimization's efficiency

Line 641: 'its' should be 'it is'

Line 711: Remove 'contain'

Fig.1: 'C.T: an' should be 'C.T: a'

Table 1: Is 'w/o' abbreviation in legend intentional?

Applicable throughtout (unsure if some of these stem from authors' typesetting in their LaTeX document or the journal's typesetting):

1) spaces in the algorithm dot notation are not being used consistently (or at least appear not to be), some have a space before the dot but not after, others have no space, some have spaces before and after the dot, etc.
This also goes for algorithm(#, #), some have no spaces after the comma and at least one has the space before the comma.

2) some S' and S" have spaces between the S and the apostrophe/inverted comma, e.g. line 401

3) there appear to be instances of double spaces after periods, although unclear if just the result of justified text

4) Three notations are used for enumeration across the document: (1), (2), etc.; (i), (ii), etc.; and (a), (b), etc. have all been used.

·

Basic reporting

This paper details a new algorithm for building the supertree used by the Open Tree of Life project. The paper is well written and clear, despite dealing with a complex algorithm.

Sufficient background is given for the context, with the code and data available. The paper is self-contained, with results that are relevant to the question being posed.

I have only two comments here. The first is figures 6, 7 and 8 need to be made clearer. It's not clear from the legend what "none" vs "batching" is compared to "none" vs "oracle". It would also be clearer to add to the caption that left is log-scaled, right is not (or just show the log-scaled?). The same applies to 7 and 8 in that I couldn't quite work out what each line meant in terms of which options were tried.

The second comment is that the * symbol used around line 200. Is that multiply? Convolution? I might also be tempted to replace the "sum" symbol with another for the set of rooted splits.

Experimental design

The manuscript is within the aims and scope and the question is well designed. Methods are described well enough to replicate.

Validity of the findings

The results show significant speed-up of the algorithm, with well-stated conclusions.

---

## Round 0.2 · accepted · Accept

Dear Dr. Redelings and Holder:

Thanks for revising your manuscript based on the concerns raised by the reviewers. I now believe that your manuscript is suitable for publication. Congratulations! I look forward to seeing this work in print, and I anticipate it being an important resource for groups studying methods for phylogeny estimation. Thanks again for choosing PeerJ to publish such important work.

Best,

-joe

Reviewer 1 ·

Basic reporting

I am satisfied with the authors changes to the original manuscript. It remains a very clear and easy to follow paper despite its inherent complexity and now makes the importance of the updated BUILD algorithm clearer to the journal's broad readership.

Experimental design

See 'Basic Reporting'.

Validity of the findings

See 'Basic Reporting'.

·

Basic reporting

The authors have responded well to all the comments by myself and the first reviewer.

Experimental design

No comment

Validity of the findings

No comment

Additional comments

No comment